# Determinants of Retroviral Integration and Implications for Gene Therapeutic MLV—Based Vectors and for a Cure for HIV-1 Infection

**DOI:** 10.3390/v15010032

**Published:** 2022-12-21

**Authors:** Eline Pellaers, Anayat Bhat, Frauke Christ, Zeger Debyser

**Affiliations:** Molecular Virology and Gene Therapy, KU Leuven, Herestraat 49, 3000 Leuven, Belgium

**Keywords:** retroviral integration, integration site selection, host factors, MLV, HIV-1, viral vector, latency

## Abstract

To complete their replication cycle, retroviruses need to integrate a DNA copy of their RNA genome into a host chromosome. Integration site selection is not random and is driven by multiple viral and cellular host factors specific to different classes of retroviruses. Today, overwhelming evidence from cell culture, animal experiments and clinical data suggests that integration sites are important for retroviral replication, oncogenesis and/or latency. In this review, we will summarize the increasing knowledge of the mechanisms underlying the integration site selection of the gammaretrovirus MLV and the lentivirus HIV-1. We will discuss how host factors of the integration site selection of retroviruses may steer the development of safer viral vectors for gene therapy. Next, we will discuss how altering the integration site preference of HIV-1 using small molecules could lead to a cure for HIV-1 infection.

## 1. Introduction

Retroviruses are divided into two main subfamilies, *Orthoretrovirinae* and *Spumaretrovirinae*. The *Orthoretrovirinae* are classified into six different genera, from *alpha* to *epsilon*, and in the genus of lentiviruses [1]. The *Spumaretrovirinae* are no longer divided into distinct genera [1]. Retroviruses typically carry their genomic information in two copies of single–stranded RNA. The retroviral replication cycle has two hallmarks: reverse transcription and integration. During reverse transcription, the viral RNA (vRNA) is copied into viral DNA (vDNA). Next, the vDNA is inserted into the DNA of the host cell, defined as integration [2].

For all retroviruses, the enzymatic process of integration, carried out by retroviral integrase (IN), is well understood. Integration of the vDNA into the host DNA is a two–step process, involving 3′ processing and strand transfer [3]. In the first step, a di–nucleotide is cleaved from both 3′ ends of the vDNA, leaving a hydroxyl at the conserved CA. After the vDNA strand enters the nucleus, a nucleophilic attack by the 3′ OH of the cleaved vDNA on the phosphodiester bridge of the host DNA is carried out. Cellular DNA repair enzymes are expected to trim the overhangs and fill the gaps, resulting in the stable integration of the provirus (reviewed in [3,4]).

Although a retrovirus can integrate its vDNA into the entire host genome, it tends to integrate with a higher frequency into specific regions; this process is referred to as integration site selection. Integration site selection is a conserved process amongst retroviruses whereby different types of retroviruses prefer distinct chromosomal sites for integration. Integration site selection has been studied for all retroviruses except for the epsilonretroviruses. For the alpharetroviruses, such as avian sarcoma–leukosis virus (ASLV), it has been reported that they have a weak preference for active genes and a nearly random integration profile [5,6,7]. For the other alpharetroviruses, such as Rous sarcoma virus (RSV), a random integration profile in the host genome has been shown [8]. The betaretroviruses, such as mouse mammary tumor virus (MMTV), display no integration site selection into specific chromatin regions or genes [9,10,11]. On the contrary, gammaretroviruses, such as murine leukemia virus (MLV), frequently integrate near transcription start sites (TSSs), CpG islands and enhancers [7,12]. The deltaretroviruses, such as human T lymphotropic virus type 1 and 2 (HTLV-1 and HTLV-2) and bovine leukemia virus (BLV), show a preferential integration pattern into transcriptional units, alphoid repetitive sequences and actively transcribed genes [13,14,15,16,17,18,19]. The lentivirus human immunodeficiency virus type 1 (HIV-1) is known to favor integration in decondensed areas of chromatin, which are associated with active transcription [20,21,22,23]. Other lentiviruses, such as feline immunodeficiency virus (FIV) and simian immunodeficiency virus (SIV), show a similar integration profile, with also a tendency to integrate near RefSeq genes [24,25]. Finally, the human foamy virus (*Spumaretrovirinae*) has been shown to integrate with a higher frequency in CpG islands and TSSs [26,27].

All studies indicate that each genus of retroviruses has a unique integration pattern in the genome of the host cell. What drives this specific integration preference of retroviruses is being revealed. Retroviral INs interact with host proteins that guide integration into preferential sites. Therefore, in this review, we will present a global picture of the determinants of integration site selection of retroviruses. We will focus on MLV and HIV-1 (Table 1), but also briefly elaborate on other retroviruses (Table 2). In the case of MLV and HIV-1, the major determinants of integration site selection are the host bromodomain and extra terminal motif (BET) proteins [28,29,30,31,32,33,34] (Figure 1) and lens epithelium–derived growth factor (LEDGF/p75) [35,36,37,38,39,40,41,42] (Figure 2), respectively. Nonetheless, the mechanism behind the integration site distribution of both retroviruses is far more complex. The study of the integration site selection of retroviruses can drive the development of safer viral vectors for gene therapy [43]. Moreover, altering the integration pattern of the HIV-1 provirus could lead to a deep latent reservoir refractory to reactivation, presenting a promising cure strategy for HIV-1 infection (reviewed in [35,44,45]).

## 2. Molecular Mechanisms Underlying MLV Integration

MLV is dependent on mitosis for nuclear entry [134,135,136]. The virus encodes a group–specific antigen (Gag) polyprotein, which is cleaved into individual proteins such as p12 [137]. The N–terminal domain (NTD) and the C–terminal domain (CTD) of p12 are important during the early stages of viral replication [138]. The NTD of p12 binds the capsid shell, resulting in a more stable capsid core, which prevents premature capsid uncoating [46,47,48]. Additionally, the CTD of p12 can bind the nucleosomes. Since p12 simultaneously binds the capsid containing the MLV pre–integration complex (PIC) and the nucleosomes, it functions as a tether of the MLV PIC to the mitotic chromosomes [48]. Indeed, a M631I mutation in the CTD of p12 interfered with the recruitment of the PIC to the nucleosomal histone proteins [48]. Furthermore, phosphorylation of p12 is known to enhance its tethering function [48]. Interestingly, the human foamy virus uses Gag instead of p12 to tether the PIC to the chromatin, also by directly binding nucleosomes [49]. Although the tethering role of p12 to mitotic chromosomes is well established, Schneider et al. reported that p12 is not a major determinant of integration site selection [50]. Replication-defective p12 mutants could be rescued by inserting alternative chromatin tethers into the mutated p12 protein. However, no significant changes in the integration sites were detected, indicating that p12 does not regulate integration site selection [50] (Table 1).

The major determinants of integration site selection are the BET proteins. During the last decade, several studies have reported that BET proteins target MLV integration towards strong enhancers, TSSs, active promotors and CpG islands [28,29,30,31,32,33,34]. All BET proteins are characterized by two conserved N-terminal tandem bromodomains (BD), BDI and BDII, and one conserved extra-terminal (ET) domain [139,140] (Figure 1A). The BDs contain hydrophobic amino acids, which interact with acetylated lysine residues on histones (Figure 1A; red domains), while the ET domain binds the MLV IN [139,140] (Figure 1A; purple domain). Through this combined interaction, BET proteins are responsible for the recruitment of the MLV IN to defined chromatin regions [139,140] (Figure 1B). Aiyer et al. showed that the CTD of MLV IN is responsible for the interaction with the ET domain of the BET proteins [51] (Figure 1A). Moreover, inducing mutations in the C-terminal 28-residue of the tail peptide of the MLV IN resulted in a shift in integration away from known BET binding sites. Strikingly, no reduction in the enzymatic activity of the MLV IN was observed with this mutation [51]. Accordingly, by the RNAi–mediated depletion of BET proteins and by using specific inhibitors targeting the bromodomain of BET proteins (such as JQ1 and I-BET), a reduction in MLV integration and the retargeting of integration sites away from TSSs was observed [29,30,31]. Of note, Acke et al. used a novel, cutting–edge technique, expansion microscopy, to study the epigenetic interactions of bromodomain–containing protein 4 (BRD4) with the acetylation marker H3K19/14Ac [52]. This method anchors the studied molecules to the polymer network of an expandable hydrogel. Due to expansion in water, physical distances within the sample are increased 4-to-5-fold, increasing the light microscopic resolution from 200 nm to 50–70 nm [52]. This study corroborated that BRD4 preferentially colocalizes with H3K19/14ac and that the addition of JQ1, a BRD4 inhibitor, results in a decrease in the colocalization of BRD4 with H3K9/24ac [52] (Table 1).

Most experiments on MLV integration site selection and the role of viral p12 and cellular BET proteins are performed in cell cultures. The question remains whether integration site selection affects the pathogenesis of MLV in its natural host. Infection of mice with MLV typically results in lymphoblastic leukemia or lymphoma [141,142]. As mentioned before, the CTD of the MLV IN—more specifically, amino acids 390–405 in the MLV IN sequence—are responsible for the interaction with BET proteins [30,51,53]. Ashkar et al. showed that the introduction of W390A in MLV IN resulted in an MLV vector characterized by an altered integration pattern in cell culture but produced at similar titers as the WT MLV [54]. Next, Nombela et al. inserted W390A as a single point mutation in the viral IN of the MLV molecular clone p63.2 [55]. By comparing wild–type (WT) MLV with BET–independent (Bin) W390A MLV replication, the role of BET proteins in MLV replication, integration and tumorigenesis was studied in vivo. First, W390A MLV replication was not significantly lower in BALB/c mice, as evidenced by similar viral loads in the spleen and thymus and by a similar number of infected cells in mice infected with WT or W390A MLV. Secondly, the effect of BET on integration site selection in spleen cells was studied 3 and 5 weeks after infection. The W390 mutation induced the retargeting of integration away from CpG islands, DNase I–hypersensitive sites, GC–enriched regions, TSSs and enhancers. Uncoupling MLV integration from BET proteins resulted in increased integration in proximity to histone marks linked with active transcription, such as H3K36me3 [55]. Next, the effect of the W390A mutation on MLV leukemogenesis in the blood, spleen and thymus was assessed. There was no statistical difference in the proliferation of white blood cells (WBCs) from WT and W390A MLV–infected mice. In addition, there was no statistical difference in the survival rates of WT compared to W390A MLV–infected mice [55]. Furthermore, an assessment of the histopathology of the spleen and thymus of mice infected with WT or W390A mutant MLV revealed no pathological differences [55]. Lastly, Nombela et al. performed a second integration site sequencing analysis on spleen cells from infected mice that had already developed lymphoma due to insertional oncogenesis. Although infection with the W390A mutant MLV still induced clonal expansion, a distinct profile was present in the integration sites with a clonal origin. The WT MLV–infected mice showed a high frequency of integration near *Myb* enhancers and the *Tmem206* and *cited2* promotors. The W390A mutant MLV displayed fewer integrations near enhancers and a higher frequency of integration into oncogene bodies such as *Notch1* and *Ppp1r16b* and promotors. Due to extensive clonal expansion, integration into these enhancers, promotors and oncogene bodies further drove the development of the lymphomas in both WT and MLV mutant–infected mice [55]. Overall these in vivo data indicate that abolishing the IN–BET interaction via a single point mutation in MLV IN can significantly alter the integration pattern by retargeting MLV integration away from enhancers, but it cannot suppress the development of lymphomas due to insertional mutagenesis [55]. A similar study was performed by the Roth group [51]. They investigated Bin MLV vectors by constructing an MLV IN mutant lacking the tail peptide of the CTD [51]. In contrast to the W390A mutant, the deletion did affect retroviral replication, confounding somewhat the interpretation of the in vivo analysis. This deletion also abolished the interaction with BET proteins and resulted in a shifted integration profile further away from TSSs and CpG islands [56]. Although integration site sequencing of Bin MLV vectors revealed reduced integration near TSSs, CpG islands and DNA hypersensitivity regions compared to BET–dependent MLV vectors, the integration pattern was not targeted away from oncogenes [51]. Loyola et al. further showed that *Myc/Runx2* transgenic mice infected with the MLV mutant lacking the tail peptide showed less tumorigenesis compared to WT MLV–infected mice, although the MLV mutant lacking the tail peptide still integrated near cancer–related genes [56]. All these studies reveal that BET is not required for retroviral replication or oncogenesis; still, they demonstrate for the first time the importance of integration site selection for retroviral pathogenesis in vivo.

All retroviral INs contain three conserved domains: the NTD, the catalytical core domain (CCD) and the CTD [3]. However, the viral IN of the gammaretroviruses contains an additional N–terminal extension (NED) [3] (Figure 1A). As mentioned before, the CTD of the gammaretroviral IN is important for the interaction with BET proteins [30,51,53]. However, Aiyer et al. suggested that both the CCD and CTD are important domains for integration site selection [58]. They corroborated, using nuclear magnetic resonance of the CTD and homology modeling of the CCD of the MLV IN, that the β_1_ and β_2_ loops of the CTD and the α_2_ helical region of the CCD mediate the binding of the IN to the target DNA [58]. After site–specific mutagenesis and motif interchanges in these domains, next–generation sequencing revealed that the integration pattern of these mutants was shifted away from TSSs and CpG islands compared to the WT, irrespective of the interaction with BET proteins [58]. Lewinski et al. further underscored the importance of the viral IN in the integration site selection of MLV [59]. They inserted the MLV IN coding region and MLV Gag into the HIV genome (referred to as HIVmGagmIN) and assessed its integration profile. Interestingly, the integration of this HIV mGagmIN resembled that of MLV, with a high frequency of integration near TSSs and CpG islands [59]. The fact that MLV IN and Gag can redirect HIV integration suggests that these may be determinants of MLV integration site selection as well [59] (Table 1).

## 3. Determinants of Integration Site Selection by Other Retroviruses

ASLV is a retrovirus belonging to the *alpha* genus, responsible for a wide range of tumors in chickens [143]. Because of its application as a viral vector for gene therapy, its integration profile has been investigated as well. Several studies claim that ASLV has a less strict integration site profile compared to HIV and MLV [5,6,7]. Nonetheless, the integration pattern is not completely random, as ASLV slightly tends to integrate near active genes [5,6,7]. Matysiak et al. proposed that the facilitates chromatin transcription (FACT) complex regulates HIV-1 integration by disassembling the nucleosomes [99]. In addition, they found that FACT binds to LEDGF/p75 and that LEDGF/p75 further stimulates the FACT–dependent promotion of HIV-1 integration [99]. Winans et al. demonstrated that the FACT complex, a histone chaperone consisting of two subunits—structure–specific recognition protein 1 (SSRP1) and suppressor of Ty16 (Spt16)—is a binding partner for ASLV [100]. Although they only proved that the FACT complex regulates the integration efficiency, it was postulated that it plays a role in integration site selection as well [100]. First, the FACT complex simultaneously binds the chromatin and the ASLV IN, similarly to other retroviral tethering factors such as BET proteins and LEDGF/p75 [100,101]. Second, the distribution of the FACT complex among the genome is related to the integration profile of ASLV [99,100,101]. Both the integrated ASLV provirus and the FACT complex are relatively randomly distributed within the genome, with a modest accumulation at transcriptionally active regions in the genome [99,100,101] (Table 2). For the other alpharetrovirus, Rous sarcoma virus (RSV), studies have reported a random integration profile of the RSV provirus in the host genome [8,102]. However, Harper et al. indicated that the CCD of the RSV IN may determine the integration site. This was evidenced by mutating the RSV integrase by inserting an alanine instead of serine at position 124 in the CCD [103]. In three assays that monitored the insertion of vDNA into non–viral DNA in vitro, the mutant RSV IN showed a distinct target preference, indicating that the CCD of the RSV IN determines the site where RSV integrates. In contrast, Shi et al. and Pandey et al. claimed that the CTD of the RSV IN is responsible for the interaction with the non–viral DNA [104,105]. Nonetheless, compared to other retroviruses, the integration profile and determinants of the integration site selection of RSV remain insufficiently studied (Table 2).

The betaretrovirus MMTV results in mammary adenocarcinomas and T–cell lymphomas in mice [106,107]. Several studies claim that these tumors are the result of insertional mutagenesis due to an integration close to oncogene bodies [107,108]. Therefore, the integration site selection of MMTV has been studied in both human and mouse cells [9,10,11]. As such, it has been corroborated that MMTV has no integration site preference into specific chromatin regions or genes [9,10,11]. Taken together, MMTV appears to have the most random distribution of integration sites among retroviruses for which this has been established to date (Table 2).

HTLV-1 is a deltaretrovirus with integration site preferences. Several studies have shown a preferential integration pattern for HTLV-1 into transcriptional units, alphoid repetitive sequences and actively transcribed genes [13,109,110,111,112,113,114,115,116,117,118]. HTLV-1 drives the development of leukemia or chronic inflammatory disease in approximately 5% of infected hosts, whereas approximately 95% remain an asymptomatic carrier [113]. However, even with the same proviral load, the degree of HTLV-1 proviral gene expression varies dramatically between infected individuals [113]. Asquith et al. demonstrated that the level of expression of HTLV-1 is correlated with the outcome of the infection. As such, a high level of gene expression is linked with an increased risk of the inflammatory conditions of the central nervous system known as HTLV-1–associated myelopathy/tropical spastic paraparesis (HAM/TSP) [113]. Furthermore, Meekings et al. revealed a role of integration site selection in the expression level of HTLV-1 and accordingly its link with the development of HAM/TSP [116]. Integration into transcriptionally active regions resulted in increased proviral gene expression, which in turn drove the development of HAM/TSP [116]. In 2015, McCallin et al. revealed a determinant of integration site selection for HTLV-1, namely the protein phosphatase 2A (PP2A) complex [119]. First, they proved with yeast–2–hybrid and co–immunoprecipitation studies that the PP2A complex interacts with HTLV-1 IN. Next, the authors revealed that PP2A knockdown was able to retarget HTLV-1 integration away from TSSs, CpG islands and epigenetic marks associated with active transcription [119]. PP2A contains four regulatory subunits. Maertens et al. further showed that the B’ regulatory subunit of PP2A is responsible for the interaction with the HTLV-1 IN [120]. The binding is genus–specific, as other *delta* viruses, such as HTLV-2 and BLV, also bind the B’ subunit of PP2A [120]. The integration pattern of HTLV-2 has been shown to be similar to HTLV-1 in vivo, with also a preference towards transcription factor and chromatin modifying binding sites such as STAT–1, p53 and histone deacetylases [14]. The third deltaretrovirus, BLV, shows frequent integration near transcription units but no association with TSSs, CpG islands or repetitive sequences [15,16,17,18,19,121]. Unlike LEDGF/p75 and BET proteins, PP2A does not directly interact with chromatin [120], raising the question of how PP2A then guides *delta* IN to specific chromatin regions. Melamed et al. showed that HTLV-1 integration sites are frequently close to binding sites for specific host transcription factors, especially STAT1, p53 and HDAC6 TFBS [118]. Instead of being targeted by a single transcription factor, HTLV-1 IN may be targeted by a protein that interacts with multiple transcription factors [144,145]. Furthermore, P22A is known to dephosphorylate several transcription factors, such as STAT1 and p53 [144,145]. Based on these observations, we could postulate that *delta* IN binds PP2A, which does not directly bind chromatin but binds transcription factors that target deltaretroviral IN to active regions in chromatin [120]. However, more research is necessary to directly test this hypothesis and search for direct binding partners. Maertens et al. further proved that PP2A stimulates the catalytic activity of the HTLV-1 IN as well [120]. Strikingly, the main determinant of HIV-1 integration site selection, LEDGF/p75, also directly stimulates the catalytic activity of the HIV-1 IN [68,69]. Hence, both PP2A and LEDGF/p75 stimulate the strand transfer activity and mediate the integration pattern of the deltaretroviral or lentiretroviral IN, respectively [68,69,120] (Table 2).

Foamy viruses, belonging to the *Spumavirinae*, also show a non–random integration pattern. Trobridge et al. investigated the integration profile of foamy viruses and found a weak preference for integration near CpG islands and TSSs [27]. The integration profile of foamy viruses is unique as it shows both clusters of integrants at certain sites of the genome and DNA gaps without integrants at other sites of the genome [27]. Interestingly, foamy viruses have no overall preference for oncogenes, indicating their potential use for gene therapy due to a reduced risk of insertional mutagenesis [27]. Nowrouzi et al. showed as well that the integration profile of foamy viruses is unique compared to other retroviruses such as MLV and HIV [130]. In addition, they claim that foamy viruses modestly prefer to integrate near promotor–close regions [130]. Beard et al. assessed the integration profile of foamy viruses compared to lentiviruses and gammaretroviruses in long–term repopulating cells from dogs [131]. They showed a preference for foamy viruses to integrate near TSSs, although to a lesser extent compared to gammaretroviruses, which reduces the likelihood of insertional mutagenesis [131]. Olszko et al. investigated the integration pattern of foamy viruses in severe combined immunodeficiency (SCID)–repopulating cells in the context of in vivo selection [132]. They showed that foamy viruses generate a polyclonal repopulation of cells without clonal dominance, which also supports the safe profile of foamy viruses for gene therapy [132]. The integration profile of foamy viruses is determined by the Gag protein [49,133]. The Gag protein contains a chromatin–binding sequence that directly interacts with the acidic patch of the nucleosomes. Interestingly, mutations in this region resulted in a shift in the integration sites of foamy viruses to centromeres [49,133]. Besides functioning as a regulating factor for integration site selection, the Gag protein also is involved in other steps of the retroviral life cycle, extending from viral trafficking to viral assembly [49,133] (Table 2).

## 4. Molecular Mechanisms Underlying HIV-1 Integration

Genomic target site selection during retroviral integration is a highly complex process, for which various biases are introduced at multiple levels [146]. The first bias appears during nuclear import. Consequently, the PIC is directed to specific chromatin regions by interacting with unique host cofactors, which further determine the integration site. Next, the viral IN itself further introduces biases in the integration site distribution [146]. Finally, the three–dimensional organization in the nucleus and the epigenetic landscape of chromatin may affect integration site selection as well [146].

The first step that mediates the integration site selection of HIV-1 is nuclear entry. Unlike gammaretroviruses, which are dependent on cell division, lentiviruses can pass through the intact nuclear envelope [134,135,136]. Numerous host factors and viral proteins are crucial for the nuclear import of the PIC, consisting of the vDNA, viral proteins and host proteins [147,148]. Examples of such factors are the ran–binding protein 2 (RANBP2, also called Nup358) [149], transportin SR2 (TRN–SR2, also called transportin 3) [150,151], nucleoporin 153 (Nup153) [149,150,151] and transportin 1 (TRN–1) [152]. Whether nuclear import plays a role in integration site selection remains a matter of debate. A link between nuclear import and the distribution of HIV-1 integration sites is expected since the nuclear entry route determines the first chromatin environment that the proviruses come across [146]. Consistently, Ocwieja et al. applied RNA interference to deplete TRN–SR2 and RanBP2, which resulted in reduced integration into gene–dense regions [60]. This was the first study to demonstrate that nuclear entry is coupled with integration site selection [60]. Moreover, Di Nunzio et al. and Lelek et al. showed that depletion of Nup153 altered the integration site preference of the HIV-1 provirus, with a reduced frequency of integration into intragenic sites [61,62]. In addition, cleavage and polyadenylation specificity factor 6 (CPSF6), a cellular protein that is proposed to interact with the viral capsid to promote nuclear entry, has been demonstrated as a determinant of HIV-1 integration site selection [63,64,65,66,67]. The role of CSPF6 in integration site selection has been studied after the depletion of CPSF6 [63,64,65,66,67] or by introducing mutations in the CPSF6 binding site of the viral capsid (N74D) [65]; in these studies, the HIV-1 integration site was targeted away from transcriptionally active chromatin regions. Nonetheless, this shift in integration pattern due to the targeting of CPSF6 did not reduce HIV-1 replication, calling into question its key role in HIV-1 replication [63,64,65,66,67] (Table 1).

Secondly, HIV-1 co–opts cellular host factors determining its integration profile. LEDGF/p75 is a member of the hepatoma–derived growth factor family and is a generally known transcriptional co–activator [70]. LEDGF is encoded by the PC4– and SFRS–interacting protein 1 (PSIP1) gene on human chromosome 9, which is expressed as two splice variants, LEDGF/p52 and LEDGF/p75. Both LEDGF/p52 and LEDGF/p75 have a similar NTD consisting of the PWWP domain (named after Pro–Trp–Trp–Pro), which interacts with H3K36me2/3, an epigenetic histone mark linked with high transcriptional activity, three charged regions (CR1, CR2 and CR3), a nuclear localization signal (NLS) and A/T hook–like elements [71]. Altogether, these regions are necessary for LEDGF/p75 to efficiently bind chromatin [71] (Figure 2A; red domains). Furthermore, LEDGF/p75, but not p52, contains a CTD that contains an integrase–binding domain (IBD) that is able to interact with the viral IN (Figure 2A; purple domain) [41,72]. Interestingly, among all lentiviruses, such as HIV-2, SIV and FIV, the interaction between IN and LEDGF/p75 is conserved [72]. With this combined interaction of the chromatin–binding domains and integrase–binding domains, LEDGF/p75 can tether HIV-1 IN to the chromatin (Figure 2B). Several studies prove that this tethering function of LEDGF/p75 is responsible for integration site selection [41]. Marshall et al. knocked down LEDGF/p75 in human cells with RNAi and they mutated the LEDGF/p75 locus in mouse cells and analyzed the integration sites of both cells [41]. The LEDGF/p75–depleted human and mouse cells showed reduced integration in transcription units [41]. In addition, Shun et al. used cells from LEDGF/p75 knockout mice to determine the integration profile in the absence of LEDGF/p75 [42]. The loss of LEDGF/p75 expression resulted in reduced integration near transcription units and more frequent integration near promotor regions and CpG islands [42]. Ciuffi et al. depleted LEDGF/p75 in 393T cells, Jurkat cells and HOS cells and analyzed their integration sites [40]. The depletion of LEDGF/p75 in all three cell lines reduced the frequency of integration near transcription units and genes regulated by LEDGF/p75 but also increased integration near GC—rich regions [40]. Furthermore, Gijsbers et al. engineered LEDGF/p75 hybrids by replacing the chromatin interaction domain of LEDGF/p75 with chromobox protein homolog 1 (CBX1), which recognizes H3K9me2/3 [38]. This resulted in the redistribution of integration while maintaining the integration efficiency [38]. This is in accordance with Meehan et al., who engineered LEDGF/p75 hybrids that had alternative chromatin tethers on the locus of the PWWP and A/T hook motifs [39]. Furthermore, Acke et al. recently used expansion microscopy to study the interaction of LEDGF/p75 with H3K36me3 [52]. The relative colocalization ratio of LEDGF/p75 with H3K36me2/3 was calculated [52]. Colocalization decreased in LEDGF/p75 knockdown cell lines [52]. In addition, the colocalization of LEDGF/p75 with H3K36me3 occurred at ~1.15 µm from the nuclear rim, consistent with the preferential area in the nucleus wherein HIV-1 provirus does integrate [52]. In addition, besides targeting the HIV-1 integration into specific genes, LEDGF/p75 enhances the catalytic activity of IN and hampers its proteolytic degradation [68,69,73] (Table 1).

Of note, the hepatoma–derived growth factor–related protein (HRP2) contains an IBD and a PWWP domain as well. As such, HRP2 can functionally substitute LEDGF/p75 after its depletion [74]. Nonetheless, LEDGF/p75 is considered the most essential factor determining integration site selection (integration site selection by LEDGF/p75 is reviewed in [45,75]). Interestingly, LEDGF/p75 has recently been implicated in post–integration latency as well [76]. During latency, LEDGF/p75 suppresses proviral transcription by recruiting Pol II–associated factor 1 (PAF1) to the proviral promotor. Intriguingly, during latency reversal, LEDGF/p75 switches its function and promotes transcription due to a casein kinase II (CKII) phosphorylation–dependent interaction with mixed lineage leukemia I protein (MLL1) and Menin, which competes with the binding of PAF1 to LEDGF/p75 [76].

As the integration of the proviral DNA in the genome of the host cell is catalyzed by the viral enzyme IN, it is not far–fetched that HIV-1 IN can direct HIV-1 integration as well. Previous studies claimed that the CCD of the HIV-1 IN was the main determinant in selecting the target sites for vDNA insertion [88,89,90]. In 2001, Harper et al. showed that a single amino acid mutation at residue 119 of HIV-1 IN resulted in an altered integration pattern [91]. Later, in 2014, Demeulemeester et al. developed IN mutants in the CCD, S119 and R231 that retargeted HIV-1 integration away from gene–dense regions [92]. Furthermore, the Parissi lab corroborated that the CTD of the HIV-1 IN contains a specific binding domain for chromatin that mediates integration site selection during the metaphase in the absence of LEDGF/p75 [93]. Other reports corroborate that the CTD can interact with histones and affect integration site selection [92,94]. The findings are further supported by the fact that the CTD of HIV-1 IN resembles a Tudor domain, which interacts with histones [93,95,96].

HIV-1 IN is post–translationally modified [97]. However, the link between post–translational modifications and integration site selection is not well studied [98]. Recently, Winans et al. defined a point mutation at an acetylation site of the CTD of the HIV-1 IN, K258, that was able to retarget the proviral integration site [98]. When this lysine at position 258 was mutated to arginine, the integration of HIV-1 was retargeted with a 10–times higher frequency to centromeric repeats as compared to the WT IN. Interestingly, these centromeric alpha satellite repeat sequences are also frequently targeted in the T cells of the latent proviral reservoir and are enriched in elite controllers (ECs) [98]. ECs are a small group of HIV–infected patients (0.2–0.5% of all HIV–infected patients) who can prevent viral rebound after discontinuation of their treatment with combination anti–retroviral therapy (cART), and they will be discussed in further detail later in this review [153]. Additionally, Winans et al. showed, with immunoprecipitation studies, that the altered integration pattern of the K258R mutation in IN is not due to differential interaction with host factors, although they did not look at the interaction with the main determinant of integration site selection, LEDGF/p75 [98]. In conclusion, it remains poorly understood which exact domains of the HIV-1 integrase are responsible for integration targeting and how post–translational modifications of HIV-1 IN affect integration site selection. Nonetheless, data suggest that, besides specific host factors, the viral IN itself plays a role in targeting integration (Table 1).

Besides nuclear import, cellular host factors and the viral IN, the three–dimensional organization in the nucleus and the epigenetic landscape of chromatin may influence integration site selection as well. Chromatin packaging affects the transcriptional state of the provirus. Heterochromatin is associated with the repression of transcription due to the highly condensed DNA that hampers the recruitment of transcriptional cofactors to the HIV promoter [154,155,156]. In euchromatin, the DNA is more open and less condensed, which makes the promoter region more accessible for transcription factors to promote transcription [154,155,156]. Furthermore, it must be mentioned that epigenetic regulation is a dynamic process. Marini et al. reported that HIV-1 prefers to integrate at a specific set of genes at the nuclear periphery, defined as HIV-1 recurrent integration genes (RIGs) [154]. In contrast, lamin–associated domains (LADS) and regions located at the center of the nucleus are less favored integration sites for HIV-1 [154]. Nuclear speckles are dynamic structures enriched in mRNA splicing factors [157]. Francis et al. showed, with quantitative imaging and integration site analysis in cell lines, primary monocyte–derived macrophages and CD4+T cells, that HIV-1 replication complexes accumulate in nuclear speckles by the use of the co–factor CPSF6. As such, nuclear speckles are a preferential integration site for HIV-1 [157]. In accordance, Rensen et al. more recently corroborated that vRNA accumulates in clusters in macrophages, which may point to the role of nuclear speckle domains in integration site selection [158]. In addition, Lucic et al. attempted to determine the genomic features of the integration site as well and claimed that HIV-1 prefers to integrate in proximity to super–enhancers [159]. Enhancers can be defined as highly acetylated genomic regions, known to easily interact with transcription factors and promotors, which in turn promote transcription [160,161]. Clusters of several enhancers are called super–enhancers and they are characterized by even higher transcriptional activity [160,161]. Interestingly, the bias in the integration pattern cannot be explained by the activity of the super–enhancers, but more likely by the three–dimensional genomic organization of super–enhancers. This further stresses the importance of analyzing the three–dimensional organization of HIV-1 integration [159].

## 5. Integration Site Selection of Other Lentiviruses

Crise et al. reported that simian immunodeficiency virus (SIV) has a similar integration pattern to HIV-1, namely favoring integration in actively transcribed regions [24]. The similar integration profile suggests that these lentiviruses have similar host factors for their integration site selection [24]. However, few studies consider in detail the exact determinants of the integration site selection of SIV. Monse et al. attempted to unravel the viral determinants of integration site selection with the use of SIV–based vectors lacking accessory proteins. Accessory proteins (Vif, Vpr, Vpx, Nef), Env and promoter or enhancer elements were excluded as essential determinants of the SIV integration pattern [128]. Derse et al. claimed that the viral integrase is probably the main viral determinant of the SIV integration profile, since closely related proviruses with similar sequences show similar integration profiles [13]. Since SIV IN also binds LEDGF/p75 and has a similar integration pattern to HIV-1, it can be deduced that LEDGF/p75 targets the integration of SIV as well [129], although no specific studies reinforce this assumption (Table 2).

Kang et al. reported that the integration profile of FIV is comparable to that of HIV and SIV but significantly different from MLV, ASLV and foamy viruses [25] (Table 2). In addition, FIV integrates near genomic regions regulated by LEDGF/p75, indicating that LEDGF/p75 targets FIV integration as well [25]. Lano et al. provided direct evidence that LEDGF/p75 is a determinant of FIV integration site selection, since the knockdown of LEDGF/p75 resulted in the redistribution of the FIV IN protein [37]. Moreover, using co–immunoprecipitation, they showed that LEDGF/p75 and the FIV IN interact and that LEDGF/p75 interacts with the FIV PIC [37] (Table 2).

HIV-1 is considered the driver of the HIV pandemic, while HIV-2 is less prevalent globally (mostly present in West Africa) and has a lower transmission rate, a slower clinical course and lower mortality [122,123]. However, when left untreated, HIV-2 can cause AIDS and eventually death, similarly to HIV-1 [122,123]. In addition, HIV-2 is characterized by a lower replication rate and a more latent phenotype compared to HIV-1, since patients infected with HIV-2 have lower RNA levels and plasma viral loads but similar DNA levels in peripheral blood molecular cells (PBMCs) [122,123,124]. For HIV-1, the higher replication rate is linked to the site of integration, since HIV-1 proviruses with higher transcriptional activity are frequently integrated into transcriptionally active regions [22]. However, it remains to be investigated whether the lower replication rate of HIV-2 is caused by a distinct integration site profile (Table 2). Unfortunately, few studies have investigated the differences in the genomic integration sites between HIV-2 and HIV-1. In 2006, MacNeil et al. investigated the integration pattern of HIV-2 in comparison to HIV-1 [125]. PBMCs were infected in vitro with an HIV-2 isolate, and the integration sites were determined. Interestingly, a significantly high proportion of both HIV-1 and HIV-2 integrated near RefSeq genes (82.2%), TSSs (16.3%), genomic regions with high GC content (44.6%) and gene–dense regions (51.2%) [125]. In addition, MacNeil et al. determined the frequency of proviral integration within heterochromatin in PBMCs from patients infected with HIV-1 or HIV-2 with an alphoid repeat PCR assay, to assess whether integration site selection is responsible for the differences between HIV-1 and HIV-2 viral replication in vivo [125]. HIV-2 showed a higher tendency to integrate within heterochromatin regions in vivo, compared to HIV-1. Therefore, although both HIV-1 and HIV-2 use LEDGF/p75 as a molecular tether, differential integration site selection may be responsible for the reduced pathogenicity of HIV-2 compared to HIV-1. However, more evidence is necessary to support this hypothesis. Besides integration site selection, other factors, such as an altered LTR promotor and distinct integration orientations, may also contribute to the differential pathogenesis of HIV-2 [126]. In 2011, Soto et al. performed an in silico study on the integration pattern in genomic DNA of PBMCs infected with either HIV-1 or HIV-2 [127]. No significant difference in the integration profile of HIV-1 versus HIV-2 was detected, since both lentiviruses integrated with high frequency near repetitive elements, CpG islands and genomic regions with high gene density [127] (Table 2).

## 6. Medical Applications of Retroviral Integration Site Selection

### 6.1. Safer Viral Vectors for Gene Therapy

Due to the unique ability of retroviruses to insert their genetic information into the DNA of the host cell, the large packaging capacity and the tissue–specific tropism, retroviral vectors are an effective gene delivery tool for a wide range of genetic diseases and some acquired diseases, such as cancer [162,163,164,165]. However, this tool comes with a risk, namely insertional mutagenesis. Insertional mutagenesis can be defined as the insertion of exogenous DNA into the genome of the host organism, which results in the deregulation of oncogenes or downregulation of tumor suppressor genes, leading to tumorigenesis [162,163,164,165]. Tumorigenesis is likely caused by clonal expansion from single cells with insertions in proximity to oncogenes [166,167,168]. In this section, we will discuss integration site selection and its link to insertional mutagenesis for two types of viral vectors: gammaretroviral and lentiviral vectors.

As mentioned before, gammaretroviral vectors tend to integrate near TSSs and enhancers [7,12], with BET proteins as the major determinant of integration site selection [28,29,30,31,32,33,34]. Due to their valuable characteristics for gene transfer, gene therapy trials have been set up with gammaretroviral vectors. Although gammaretroviral vectors proved effective, major concerns about safety emerged when two of the ten children, who were supposed to be treated for X–linked severe combined immunodeficiency (SCID–X1) syndrome with MLV vectors, developed T cell leukemia due to an integration near the *LMO2* oncogene [169,170]. In addition, in a gene therapy trial for Wiskot Aldrich syndrome, gammaretroviral vectors were found to integrate near *LMO2*, *MDS1*, *MN1*, *CCND2*, *BMI1* and *EVI1*, all causing mutagenesis [171,172]. Moreover, the treatment of chronic granulomatous disease with gammaretroviral vectors also resulted in the activation of the *EVI1* gene [173]. However, for other diseases, such as adenosine deaminase (ADA)–SCID, no insertional mutagenesis has been reported so far [163,174]. Some studies doubt that the reported tumorigenesis of MLV vectors is the result of insertional mutagenesis, but claim that it is rather caused by the gene product that is inserted in the MLV vector—more specifically, interleukin 2 receptor subunit gamma (ILRG2) in the case of X–SCID gene therapy [175,176]. However, after these incidents, several attempts have been made to improve the biosafety level of viral vectors. First, self–inactivating (SIN) gammaretroviral vectors, which carry a deletion in the LTR U3 sequence encoding for enhancer and promotor functions, have been developed to decrease the probability of oncogene activation [177,178,179]. Indeed, clinical trials have shown that although the global integration pattern of second–generation SIN and first–generation MLV vectors is similar, adverse events due to integration near proto–oncogenes are reduced [180].

The Bushman lab was the first to propose to control the integration site selection of retroviruses to increase the safety profile of viral vectors for gene therapy by fusing the retroviral integrase to DNA–binding domains [181]. A more recent effort was based on the W390A mutation in the MLV IN, as described above. The W390A mutation abolishes the interaction of the MLV IN with BET proteins [55,57]. This results in a distinct integration profile with less integration in oncogenic TSSs [55,57]. As discussed, Nombela et al. further confirmed the altered integration pattern of the BET–independent (Bin) MLV virus in vivo [55]. However, retargeting was not sufficient to prevent lymphomagenesis due to insertional oncogenesis [55]. The study of the Roth group gave similar results; BET is not required for oncogenesis in mice. However, it should be noted that both studies investigated the oncogenic potential of replication–competent BET–independent retroviruses, not BET–independent retroviral vectors [56].

Adverse events with gammaretroviral vectors have shifted the focus of researchers toward lentiviral vectors. Lentiviral vectors prefer to integrate near actively transcribed genes [20,21,22,23]. LEDGF/p75 can be considered as the ‘GPS’ of the provirus to target the integration to these sites [22,35,36,73,75,78]. Although this integration pattern seems to be safer, lentiviral vectors come also with risks, as aberrant splicing [178,182] and clonal expansion [183] have been observed. As a result, SIN lentiviral vectors have been developed, similar to SIN MLV vectors [180]. Cesana et al. further assessed the genotoxicity of SIN lentiviral vectors and confirmed that although there was less integration near oncogenes, residual genotoxicity was still observed due to the inactivation of tumor suppressor genes [178]. Altering the integration of lentiviral vectors was also adopted as an approach to increase safety. Vranckx et al. generated LEDGF hybrids by replacing the PWWP domain of LEDGF with pan–chromatin binders, which resulted in more random integration within the host cell genome while maintaining stable transgene expression [79]. Lapaillerie et al. further proved that the CTD of the HIV-1 IN has an affinity for specific chromatin regions, mediating HIV-1 integration site selection, in the absence of LEDGF/p75. In the presence of LEDGF/p75, the chromatin–binding properties of HIV-1 IN change, resulting in a distinct insertion pattern [93]. Furthermore, Benleulmi et al. demonstrated that the CTD of HIV-1 IN interacts with the amino–terminal peptide tail of histone 4. Accordingly, mutants defective for interaction between the HIV-1 IN and histone 4 show less efficient integration and a shift in integration towards less condensed dynamic chromatin regions. Mutant HIV-1 IN still interacted with LEDGF/p75, indicating that the abolished interaction of HIV-1 IN with histone 4 is responsible for retargeting the HIV-1 integration [94]. Altogether, retroviral vectors represent a promising and effective tool for gene therapy, but increased insight into the integration site selection process of gamma– and lentiviruses is warranted to further improve the safety profile of these vectors.

Viral vectors can also be derived from other genera of retroviruses, such as alphaviruses. Due to the more random integration profile of alpharetroviruses [5,6,7,8,102], a safer profile for alpharetrovirus–based vector delivery systems has been predicted, which has resulted in the development of SIN alpharetrovirus–derived vectors [6,184,185,186,187,188,189,190]. These vectors have no preferential targeting for TSSs, transcribed genes, CpG islands, conserved non–coding regions, elements enriched in transcription factor binding sites, promotor regions, enhancer regions, repetitive elements and, most importantly, oncogene bodies [6]. According to Kaufmann et al., no aberrant splicing occurs with these vector systems [191]. Nonetheless, the integration profile and determinants of the integration profile of alpharetroviruses are less studied compared to gamma– and lentiviruses.

In addition, foamy viruses have already been implicated in gene therapy for a long time [184]. Foamy virus vectors have several beneficial characteristics, such as their wide tropism [192], the non–pathogenic nature of the parental virus [193] and their large transgene capacity (12–13 kb) [194]. Moreover, they can be produced easily at high titers [195]. Therefore, foamy virus–derived vectors have been tested in preclinical studies for a wide range of diseases, such as SCID–X1 [196,197,198], leukocyte adhesion deficiency type 1 [199], Duchene muscular dystrophy [200] and hematopoietic stem cell therapy [131,132,195,196,199,201,202,203]. In terms of their safety profile, foamy virus viral vectors show a relatively safe integration pattern [202,204]. Everson et al. compared the safety profile of foamy viral vectors with that of lentiviral vectors and corroborated that foamy viral vectors integrate less near RefSeq genes, proto–oncogenes and TSSs [203]. Although foamy vector–transduced cells were more polyclonal, they showed less dominant groups, with no integrations near proto–oncogenes in the few dominant clones that were present [203]. Three approaches have been used to further enhance the safety profile of foamy viral vectors: inserting housekeeping promotors, inserting insulators and retargeting foamy retroviral integration [205]. Viral promotors, such as that derived from spleen focus forming virus (SFFV), are known to cause clonal expansion and malignancy via the strong activation of nearby genes [178,205]. Therefore, housekeeping promotors such as elongation factor 1–α (EF1–α) or phosphoglycerate kinase (PGK) are used instead [206]. As a second approach, insulators can be incorporated, which can inhibit the interaction between the enhancer and promotor [202]. Retargeting the integration profile was considered as a third approach by Hocum et al. [26]. Foamy viral integration was retargeted away from genes and proto–oncogenes and retargeted more towards satellite regions enriched with H3K9me3 [26]. They achieved this integration profile by modifying the Gag and Pol protein of the foamy virus plasmids. More specifically, the Pol construct expressed CBX1 fused to the FV IN, while the Gag construct carried mutations in the chromatin–binding site to reduce the binding affinity of Gag to the host chromatin [26]. In conclusion, foamy virus–derived vectors represent a promising alternative gene delivery tool for human gene therapy.

### 6.2. Towards a Cure for HIV-1 Infection

Significant advances in HIV treatment have resulted in the increased life expectancy of people living with HIV (PLWH) [207]. Combination antiretroviral therapy (cART) reduces the viral load by inhibiting HIV replication, allowing the immune system to recover and even preventing transmission to uninfected individuals [207,208]. According to the World Health Organization, 28.7 million people were treated with cART in 2021, which covers only 75% of all PLWH [207]. Additionally, cART is associated with numerous adverse effects (e.g., central fat accumulation, renal toxic effects, liver toxicity, hypersensitivity reactions and osteoporosis) [209]. From an economical point of view, the cost of cART therapy is not sustainable [210]. Furthermore, the treatment requires strict, lifelong adherence to impede the emergence of drug–resistant strains [211,212,213]. HIV is known to integrate into the genome of the cluster of differentiation 4 (CD4) + host cell. Some of the host cells are long–lived memory CD4+ cells, where the virus resides in a latent state, not recognized by the immune system or susceptible to cART [208,214]. As a result, cART cannot cure HIV infection, and treatment discontinuation results in a viral rebound as the latently infected cells carry the replication–competent provirus that is activated together with the memory CD4 cell [208]. Consequently, lifelong therapy is required [208]. To develop novel approaches and targets to cure HIV-1 infection, we must continue to expand our basic understanding of the molecular virology of HIV.

To date, most dominant HIV-1 cure strategies aim to eliminate the latent reservoir, and most efforts focus on the so–called shock–and–kill cure strategy (reviewed in [215]). The shock–and–kill strategy represents a sterilizing cure strategy aimed at the complete eradication of HIV-1 via the reactivation of latently infected cells. During the shock phase, the production of viral proteins is stimulated through reactivation of the latent reservoir with latency–reversing agents (LRAs). Later, during the kill phase, the virus is eliminated by immune–mediated clearance or by the cytopathic effect of the reactivated virus [215]. Despite extensive research, this approach must overcome several challenges before being translated into the clinic. First, LRAs may have life–threatening adverse effects, such as the induction of a cytokine storm [215]. Secondly, most LRAs only target CD4+T cells, disregarding other cell types that also constitute the latent reservoir, such as macrophages [215,216]. Reinfection can also occur through the activation of non–infected HIV-1 target cells. Moreover, although LRAs induce viral RNA (vRNA) transcription, and they fail to reduce the size of the latent reservoir in clinical trials [217,218,219]. The inability to reduce the reservoir size is probably the result of the diverse nature of the latent reservoir and the multiple pathways that regulate HIV-1 gene expression. Therefore, a profound understanding of the regulatory mechanisms of HIV-1 silencing, and the discovery of safe and effective LRAs, is of paramount importance to increase the efficiency of the shock–and–kill strategy. Chen et al. suggested that a further understanding of integration site selection is important for an efficient shock–and–kill strategy [220]. They showed that HIV transcription is stimulated by integration in proximity to enhancer regions and that the latent provirus is integrated at an increased distance from enhancers [220]. In addition, they corroborated that the response to LRAs is linked to their integration site. For example, two LRAs, phytohemagglutinin and vorinostat, can reactivate different subsets of proviruses integrated at different positions in the genome [220]. This further underscores that increased insight into integration site selection and its link to transcription is critical for the development of LRAs. Similarly, Battivelli et al. highlighted the importance of integration site selection for the shock–and–kill strategy by pinpointing the difference in the integration sites between cell populations that are sensitive or refractory to reactivation [221].

The limited success of the shock–and–kill strategy resulted in the investigation of alternative approaches, such as a block–and–lock functional cure strategy (reviewed in [222,223,224]). With the use of latency–promoting agents (LPAs), the block–and–lock cure strategy strives to generate a cellular reservoir that is resistant to reactivation and unable to rebound after treatment interruption [222,223,224]. Several LPAs have been described, such as the Tat inhibitor didehydro–cortistatin A [225,226], curaxin CBL0100 [227], heat shock protein 90 (HSP90) inhibitors [228], Janus kinase signal transducer and activator of transcription (JAK–STAT) pathway inhibitors [229], mammalian target of rapamycin signaling inhibitors [230] and LEDGINs [22,23,77] (for a review, see [223,231]). In this review on retroviral integration, we will further discuss the block–and–lock functional cure with LEDGINs.

LEDGINs are small molecules targeting the LEDGF/p75 binding pocket of the HIV-1 IN [80]. Initially, the LEDGF/p75–IN interaction was considered a therapeutic target for antiretroviral therapy [81] because both the knockdown of LEDGF/p75 [82] and overexpression of the IBD of LEDGF/p75 [83] resulted in the potent inhibition of HIV replication in cell culture. LEDGINs display both early and late effects on the viral replication cycle. LEDGINs allosterically inhibit the catalytic activity of IN and inhibit HIV-1 integration [69,84,85]. As for the late effect, LEDGINs stimulate IN oligomerization prematurely, leading to defective progeny virions [85,86]. The ribonucleoprotein of these defective viral particles is located outside the capsid core, whereas other particles do not even have a core [69,85,86]. Moreover, the particles formed in the presence of LEDGINs are characterized by less efficient reverse transcription, nuclear import and integration [69,85,86]. Later on, the Debyser lab showed the potential of LEDGINs in a block–and–lock cure strategy. Vranckx et al. showed that LEDGINs retarget HIV-1 integration out of transcriptionally active regions and shift integration towards the inner nuclear compartment. The transcriptional state of the residual provirus proved to be reduced and refractory to reactivation. These results were reproducible in primary CD4+T cells [77]. Vansant et al. further investigated the late effect of LEDGINs [87]. They corroborated that when LEDGINs are present during virus production, the integration and expression patterns of the residual provirus are affected, resulting in a more quiescent provirus [87]. The integration pattern relocated at an increased distance from active chromatin regions, such as DNase I–hypersensitive sites, CpG islands, GC–rich regions and active transcription markers. However, the integration near H3K36me3 and active genes was not altered. These results were confirmed in primary cells as well [87]. In 2020, employing the barcoded HIV–ensembles (B–HIVE) technology, which uses a unique barcode to tag the HIV genome in order to trace insert–specific HIV expression, the underlying epigenetic mechanism of the LEDGIN–induced block–and–lock phenotype was revealed [22]. This research showed that LEDGIN treatment enhanced the distance of integration to H3K36me3, the marker recognized by LEDGF/p75; decreased the viral RNA (vRNA) expression per residual vDNA copy; and enlarged the proportion of silent provirus [22]. Interestingly, these results also showed that LEDGINs do not influence the proximity of the integration site to enhancers [22]. Most recently, Janssens et al. underscored again the proof–of–concept for LEDGINs in a block–and–lock cure strategy with branched–DNA (bDNA) imaging. This technique can simultaneously quantify the vDNA and vRNA spots in single cells in HIV latency cell line models and primary cells [23]. With this approach, Janssens et al. investigated how LEDGIN treatment (CX014442) influences the three-dimensional location, basal transcription and TNF—α—mediated reactivation of HIV-1 compared to raltegravir, a clinically approved integrase strand transfer inhibitor [23]. Treatment with LEDGINs and raltegravir dose—dependently decreased the vDNA level, indicating that they both inhibit HIV-1 integration [23]. Raltegravir treatment only modestly decreased the number of vRNA spots per infected cell in both unreactivated and TNF—α—treated cells. On the contrary, the addition of LEDGINs potently reduced HIV-1 basal transcription and reactivation from latency in a dose–dependent manner. Since LEDGINs reduce the vRNA level per vDNA copy, the observed phenotype is not solely due to reduced integration [23]. Because bDNA imaging provides spatial information, the minimum distance between the vDNA spot and the nuclear boundary could be calculated. This calculation evidenced that treatment with LEDGINs but not with raltegravir shifted the spatial location of the integrated provirus further away from the nuclear boundary [23]. Additionally, GS–9822, a LEDGIN congener with activity in the nanomolar range, was investigated. The addition of GS–9822 hampered HIV-1 transcription and reactivation, even at low nanomolar concentrations [23]. Finally, to investigate the effect of LEDGINs in a more translational setting, PBMCs and CD4+T cells were treated with LEDGINs. In primary cells as well, treatment with LEDGINs inhibited the basal transcription and reactivation of the provirus, as evidenced by the reduction in vRNA expression per residual vDNA copy. Interestingly, measurement of the integrated copies with Alu–LTR qPCR over time corroborated that the treatment of primary cells with LEDGINs induced provirus enrichment in a deep latent state [23]. Overall, these results confirm the impact of integration site selection on the HIV-1 transcriptional state. LEDGINs can permanently silence HIV-1 expression due to retargeting the integration of the provirus.

The clinical importance of integration site selection and the validity of a LEDGIN–based future block-and-lock cure approach was emphasized by the Lichterfeld group in a study on a specific group of patients, called ECs [153]. As mentioned before, this small group of patients (0.2–0.5% of all HIV–infected patients) can prevent viral rebound after the discontinuation of cART. Jiang et al. demonstrated that the latent viral reservoirs of ECs are in a state of long-lasting deep latency, which is linked to their integration site [153]. Proviruses of ECs seem to be more frequently integrated in non-protein-coding regions and in proximity to the densely packed centrum of a chromosome. In addition, ECs have a preferred integration pattern in genes that encode members of the zinc-finger protein family, associated with strongly repressed transcription [153]. The authors proposed that the deep latent reservoir of ECs might be explained by the cell–mediated immune killing of the transcriptionally competent provirus over time, rather than favored integration in transcriptionally silent regions [153]. This means that proviral DNA that is integrated into transcriptionally active regions of the genome is eliminated by the immune system over time, while a deep latent reservoir is selected. This hypothesis is supported by the observation that infected cells from ECs and people treated with cART have a similar integration pattern in vitro, and by the fact that ECs have an unusually potent immune response against HIV–infected cells [153]. Later, the Lichterfeld lab provided additional evidence for the importance of integration sites in a block-and-lock strategy through a second observation. Einkauf et al. were the first to study the link between the proviral sequence and the integration site in patient samples after cART, using matched integration site and proviral sequencing (MIP–seq) [232]. After long–term cART, the majority of intact proviruses were located in non–genic, transcriptionally silent regions, and integrated in opposite orientation to the host gene, compared to defective proviruses [232]. Further, RNA sequencing was used to calculate the chromosomal distance between each proviral integration site and the most distant TSS, and to evaluate their transcriptional activity [232]. In addition, the application of transposase–accessible chromatin using sequencing (ATAC–seq) was used to determine the chromosomal accessibility of the genomic DNA regions. This corroborated that there was a selection of intact proviruses located in less accessible chromatin regions with an increased distance to active TSSs, associated with deeper levels of latency [232]. However, the authors also noticed that, besides the selection advantage of transcriptionally silent proviruses, some transcriptionally active clones persist despite long–term cART, representing a “loud minority” [232]. Briefly, Einkauf et al. suggested that long–term cART is associated with a shift in the composition of the viral reservoir, likely causing the accumulation of intact proviral sequences with progressively increasing depths of viral latency [232]. Long–term cART may thus promote a profile of a deep latent reservoir resistant to reactivation, resembling the reservoir obtained after a block–and–lock cure approach. As such, the Lichterfeld lab provided clinical evidence to support a future block–and–lock cure strategy with both elite controllers [153] and patients under long–term cART [232].

### 6.3. Optimizing Oncovirotherapy

An efficient therapy for cancer remains a huge unmet medical need [233]. Recently, oncolytic virotherapy (OVT) emerged as an innovative therapeutic strategy. It utilizes replicational–competent viruses to selectively kill tumor cells (reviewed in [234,235]). Oncolytic viruses have multimodal anticancer activity as they selectively target the tumor, wherein they replicate and induce a general antitumor immune response [234,235]. Oncolytic viruses have proven their potency as an anticancer agent in many clinical trials [236]. Diverse viruses have been applied for OVT, such as adenoviruses, herpesviruses, vaccinia viruses, measles viruses, polio viruses, Newcastle disease viruses, coxsackieviruses, reoviruses, parvoviruses H1, vesicular stomatitis virus and novel nano–pseudovirus [234,235,236]. Interestingly for this review, retroviruses are also applicable for OVT [236]. The MLV virus is a promising oncolytic virus as it is well investigated, it replicates solely in dividing cells—thus increasing the selectivity for tumor cells—and, as anti–retroviral drugs are available to stop viral replication, it alleviates safety concerns [237]. Although MLV does not immediately induce cell lysis or immune activation, it shows stable integration and can carry suicide genes to selectively target the host cells that. are infected [237]. MLV–based oncolytic vectors have been proposed for human osteosarcoma [238] or malignant mesothelioma [239]. A lead clinical candidate among MLV–replicating vectors for OVT is ‘Toca 511’, which is an engineered MLV virus, encoding yeast cytosine deaminase. This enzyme can convert 5–fluorocytosine to the toxic anticancer agent 5–fluorouracil in the tumor cell, wherein this virus replicates [240,241]. Animal testing indicates that this vector shows high selectivity for tumor cells and has, as such, a promising safety profile [240,241]. Toca511 has therefore progressed into phase III clinical trials for malignant glioma [238]. However, one of the major hurdles for MLV–based oncolytic viruses is their short intracellular half–life [242]. The intracellular half–life of MLV is only 5.5 to 7.5 h, and MLV can only integrate during cell division, when the nuclear membrane is disassembled [242]. This restricts the application of MLV for OVT as the suicide genes are not spread slowly in the growing tumor [242]. A retroviral alternative could be the foamy viruses. As with MLV, they only infect dividing cells. In contrast to MLV, their intracellular half–life is 30 days, allowing a gradual infection in slowly dividing tumor cells. Other beneficial characteristics are their non–pathogenicity, their ability to destroy cells through syncytium formation, the low seroprevalence in humans and their random integration pattern, which reduces the risk of insertional mutagenesis [241,242]. Budzik et al. further stressed the potential of foamy vectors for OVT by designing oncolytic foamy vectors with recombinant chimeric chimpanzee simian foamy viruses engineered with suicide transgenes [243]. These vectors can successfully infect multiple human cancer cells with slow kinetics, which resulted in growth arrest and prolonged survival in animal models [243,244]. In 2022, Budzik generated foamy viral oncolytic vectors carrying thymidine kinase and inducible caspase 9 [245]. These vectors are able to stably carry and express transgenes upon serial passage, suggesting foamy viruses as a promising oncolytic virus for OVT [245]. Nonetheless, a better understanding of the strategies developed by retroviruses to ensure efficient integration and replication will be of great interest in order to engineer and optimize future OVT applications.

## 7. Retroviral Integration Sites Matter

It is well established now that the distribution of retroviral integrants is determined by several viral and host factors. In brief, the MLV PIC depends on the breakdown of the nuclear envelope during mitosis to enter the nucleus. P12 tethers the MLV PIC to the chromosomes, although its role in integration site selection is under debate [46,47,48,49,50]. However, it is clear from numerous reports that BET proteins determine the integration site selection of the MLV PIC near enhancers, TSSs and CpG islands [28,29,30,31,32,33,34] (Figure 3; left panel). Lentiviruses, on the other hand, have an active nuclear entry route, as they can cross nuclear pore complexes. Several host factors involved in nuclear import affect integration site selection, such as TRN–SR–2 [60] and CPSF6 [63,64,65,66,67]. The nuclear pore proteins RanBP2 [60] and Nup153 [61,62] are also claimed to be involved in regulating the integration profile of HIV-1. The chromatin reader LEDGF/p75 is considered the main director of HIV-1 integration as it tethers the viral IN to the actively transcribed regions of chromatin [38,39,40,41,42,52]. Additionally, the three–dimensional and epigenetic landscape of the chromatin environment determines HIV-1 integration and the transcriptional state of the provirus [154,155,156,159]. HIV-1 is more likely to integrate into less condensed euchromatin, which is associated with active transcription, and less likely to integrate into closely packed heterochromatin, which is associated with repressed transcriptional activity [154,155,156,159] (Figure 3; right panel).

By interference with the two major determinants of integration site selection, BET proteins for MLV and LEDGF/p75 for HIV-1, integration can be retargeted out of the naturally preferred regions. BET inhibitors such as JQ1 and I–BET have been claimed to retarget the integration away from TSSs [29,30,31]. A clinical application of this insight is the development of Bin MLV vectors, which have been used to reduce the risk of insertional mutagenesis [55,56,57]. Similarly, retargeting HIV-1 integration by intervening with LEDGF/p75 [79] or with the CTD of HIV-1 IN [93,94] also reduced the risk of insertional mutagenesis for HIV-1 viral vectors. Moreover, by inhibiting the interaction between HIV-1 IN and LEDGF/p75 with small molecules named LEDGINs, HIV-1 integration can be retargeted away from H3K36me2/3, the marker recognized by LEDGF/p75 [22,23,77]. As such, LEDGINs are used to retarget integration into silent regions of the genome, resulting in a reservoir refractory to reactivation [22,23,77]. To conclude, LEDGINs provide us with a block–and–lock cure strategy that aims to permanently silence the latent reservoir, even after treatment interruption [22,23,77].

## 8. Perspectives and Open Research Questions

### 8.1. Safer Viral Vectors for Gene Therapy

The integration pattern of MLV has been widely investigated and BET proteins are recognized as the GPS for integration targeting. When uncoupling the interaction of BET proteins and MLV IN, integration is retargeted. Interestingly, Aiyer et al. indicated that no reduction in the enzymatic activity of the MLV IN was observed when uncoupling the interaction with BET proteins [51]. Furthermore, Nombela et al. demonstrated that neither viral replication nor integration in vivo can be prevented with the W390A mutant, which abolishes the interaction of BET proteins and the viral IN [55]. This implies that other host factors, chromatin features or the viral IN itself mediate the integration of the W390A mutant [55].

For MLV, p12 serves as a tether of the MLV PIC to chromosomes [46,47,48,49,50]. Investigation into the relative roles of p12 and BET proteins in integration site selection is warranted. In addition, Nombela et al. corroborated that Bin MLV still drives the same disease phenotype as BET–dependent MLV. In addition, they also claim that abolishing the BET–IN interaction cannot suppress the development of lymphomas due to insertional mutagenesis, indicating that the interaction of MLV and BET proteins is not crucial for oncogenesis [55]. It remains unknown how the Bin MLV virus generates lymphoma. Nombela et al. postulated that WT and W390A–induced lymphomas result from a different mechanism [55]. As WT MLV tends to integrate in proximity to promotors and enhancers, lymphomagenesis is probably provoked by integration near enhancers and promotors, which results in the overexpression of cellular oncogenes. In contrast, the W390A mutant directly integrates near oncogene bodies, resulting in insertional mutagenesis [55,57] (Figure 4). Nonetheless, these experiments were performed with replication–competent retroviruses, and it is not clear whether we can extrapolate these findings to single–round MLV vectors. Although Bin MLV–based vectors and viruses do alter the integration profile out of enhancer regions [55,56,57], this change is not sufficient to create a safe toxicity profile for the retrovirus in vivo. However, cellular enhancers may drive oncogenesis in WT MLV and W390A MLV vectors. SIN vectors, without the retroviral enhancer/promotor, do reduce the insertional mutagenesis of Bin vectors even further. Therefore, an interesting approach would be to combine the principles of SIN and Bin MLV vectors to further enhance the safety profile of MLV vectors. However, in vivo experiments with Bin SIN MLV vectors are required to assess the genotoxicity profiles of such third–generation vectors.

### 8.2. Towards an HIV-1 Cure

The experimental results on the LEDGIN–mediated retargeting of integration are positive and point toward the use of LEDGINs in a block-and-lock cure strategy, although this approach still faces some key challenges. Before moving to the clinic, more advanced studies with LEDGINs should be conducted in humanized mice and patients. Furthermore, it remains to be determined how LEDGINs could be applied in clinical settings to HIV-1—infected patients. LEDGINs could be added to Pre–Exposure Prophylaxis (PrEP) to silence residual proviruses that escape PrEP [44]. Because LEDGINs inhibit integration, they should be administered as soon as possible during acute infection to interfere with the formation of the latent reservoir. In addition, as early treatment has been shown to reduce the size of the latent reservoir [44,246] and as LEDGINs produce a latent reservoir refractory to latency reversal [22,23,77], LEDGINs may quantitatively and qualitatively reshape the latent reservoir when added during early treatment. Furthermore, the Swanstrom group has shown that the replication–competent reservoir is primarily established near the time of therapy initiation [214,247], pointing towards the potential role of LEDGINs in patients diagnosed years after infection. One of the main hurdles is how chronically infected patients that are treated with cART for a long time can be treated with LEDGINs. It is unknown how LEDGINs can affect the dynamics of a latent reservoir that is already established. The most interesting approach would be to set up clinical trials wherein patients undergo treatment interruption, whereafter LEDGINs are administered in combination with cART [44]. An estimation of the reservoir size with a quantitative viral outgrowth assay (qVOA) and proviral DNA loads for these patients could provide interesting information about the clinical potential of LEDGINs [44]. Of course, it is not known yet to which extent the latent reservoir can be reactivated and what the effect of LEDGINs is in this context.

Considering the challenges of the heterogenous complexity of HIV-1 transcription and the diverse integration landscape of HIV-1 proviruses, it may be crucial to establish an optimal ‘pro–reactivating combinatorial cocktail’ for the shock–and–kill strategy, or a ‘pro–latency combinatorial cocktail’ to increase the efficiency of the block–and–lock approach in a more translational setting. As Vansant et al. proved that LEDGINs do not influence the proximity of integration near enhancers [22], and as enhancers are known to promote transcription [160,161], the residual RNA expressing proviruses after LEDGIN treatment are likely caused by stochastic integration near enhancer regions. Therefore, enhancer antagonists could be combined with LEDGINs to increase the efficiency of the block–and–lock phenotype. Other LPAs with independent mechanisms could be combined with LEDGINs as well. A promising example is a Tat inhibitor, didehydro–cortistatin, that has been shown to have latency–promoting activity in mouse models [225,226].

Another interesting yet uninvestigated option could be the combination of both, in a ‘lock–and–shock’ approach. As Janssens et al. [23] suggested, we could first apply the block–and–lock cure approach to silence the latent reservoir and afterward eradicate the residual high–expressing provirus with the shock–and–kill strategy. Therefore, first, a ‘pro–latency combinatorial cocktail’ with LEDGINs could be added during the early treatment of patients to diminish the size and functionality of the functional reservoir. Subsequently, a combination of different LRAs can be given to the patient to reactivate the residual replicational–competent provirus, followed by immune–mediated eradication [23]. However, even with this strategy, the diverse nature of HIV-1 integration and transcription remains a major challenge, and further investigation into the determinants of HIV-1 integration site selection and transcription remains warranted.

## Figures and Tables

**Figure 1 viruses-15-00032-f001:**
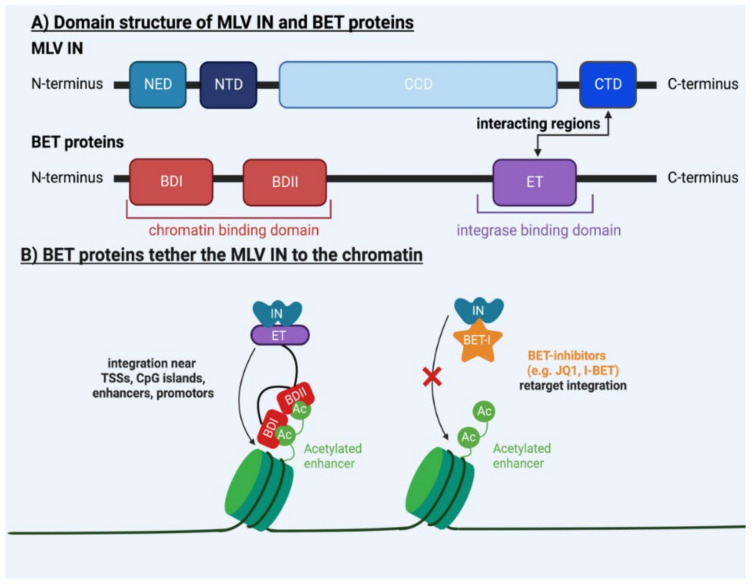
(**A**) Domain structure of murine leukemia virus (MLV) integrase (IN) and bromodomain and extra–terminal motif (BET) proteins. The MLV IN consists of an N–terminal extension domain (NED), an N–terminal domain (NTD), a catalytic core domain (CCD) and a C–terminal domain (CTD) [4]. BET proteins consist of two bromodomains (BD), BDI and BDII, that bind acetylated lysines in the nucleosomes, and an extra–terminal (ET) domain that interacts with the CTD of the MLV IN [28,29,30,31,32,33,34]. (**B**) BET proteins tether the MLV IN. Through the combined interaction of the ET domain of BET proteins (which binds the CTD of the MLV IN) and BDI and BDII (which bind the acetylated chromatin), BET proteins tether the MLV IN to acetylated chromatin regions such as enhancers [28,29,30,31,32,33,34]. BET inhibitors, such as JQ1 and I–BET, retarget the integration of MLV away from BET–recognized sites by uncoupling the interaction between BET proteins and MLV IN [29,30,31]. (Figure created with Biorender.com on 17 December 2022).

**Figure 2 viruses-15-00032-f002:**
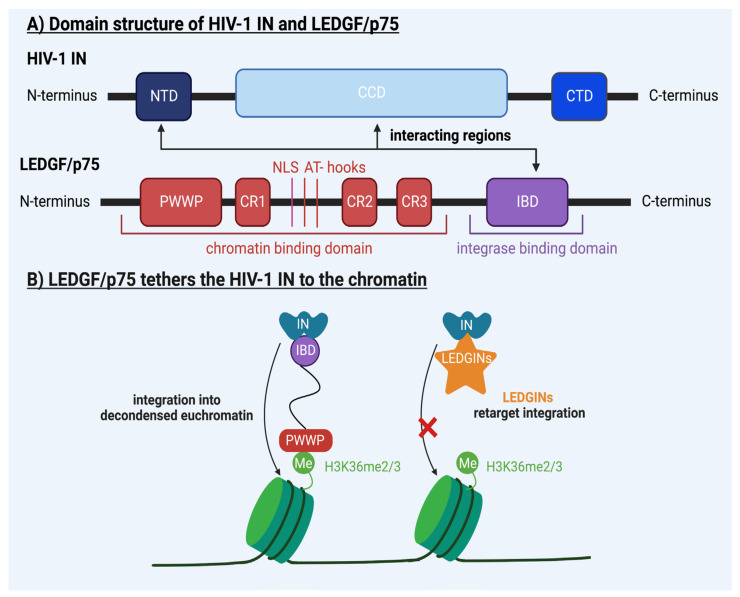
(**A**) Domain structure of HIV-1 integrase (IN) and lens epithelium–derived growth factor (LEDGF/p75). HIV-1 IN consists of an N–terminal domain (NTD), a catalytic core domain (CCD) and a C–terminal domain (CTD) [3]. LEDGF/p75 consists of an integrase–binding domain (IBD), binding the viral IN, and a PWWP domain, binding epigenetic marks in the genome associated with active transcription such as H3K36me2/3 [38,39,40,41,42]. (**B**) LEDGF/p75 tethers HIV-1 IN to chromatin. LEDGF/p75 simultaneously binds HIV-1 IN with its integrase–binding domain (IBD) and chromatin through its PWWP domain, tethering HIV-1 IN to decondensed chromatin regions, tagged by H3K36me2/3 [38,39,40,41,42]. HIV-1 integration can be targeted away from H3K36me2/3 with the use of LEDGINs, small molecules that inhibit the interaction between HIV-1 IN and LEDGF/p75 [22,23]. (Figure created with Biorender.com on 17 December 2022).

**Figure 3 viruses-15-00032-f003:**
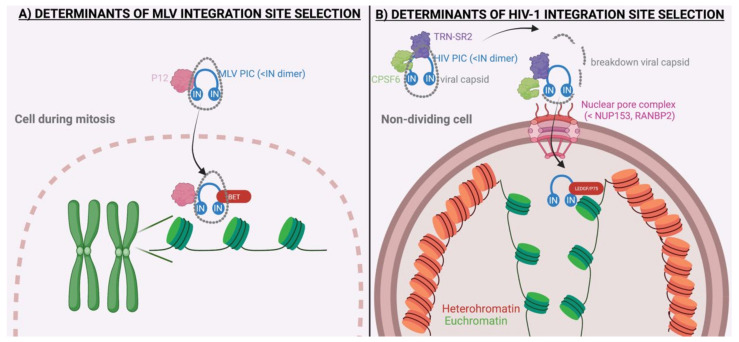
(**A**) Determinants of the murine leukemia virus (MLV). MLV tends to integrate near transcription start sites (TSSs) and enhancers. The MLV p12 functions as a tether of the pre–integration complex (PIC) to the chromosomes, but its role in integration site selection is under debate. In contrast to HIV-1, MLV does not have an active nuclear entry route but is dependent on mitosis for nuclear entry. After nuclear entry and mitosis, bromodomain and extra–terminal motif (BET) proteins direct the integration pattern of MLV. BET proteins consist of an extra–terminal domain, which binds the MLV integrase (IN), and two bromodomains (BDI and BDII) that bin acetylated lysine in the nucleosomes. (**B**) Determinants of human immunodeficiency virus 1 (HIV-1). HIV-1 tends to integrate near decondensed chromatin, associated with active transcription, at the nuclear periphery. The HIV-1 pre–integration complex (PIC) can infect non–dividing cells by crossing the intact nuclear membrane via nuclear pore complexes. Nuclear import affects the HIV-1 integration pattern, with the ran–binding protein 2 (RANBP2, also called Nup358), transportin SR2 (TRN–SR2, also called transportin 3) and nucleoporin 153 (Nup153) as determinants of integration site selection. After nuclear import, lens epithelium–derived growth factor (LEDGF/p75) is considered the major contributing factor to integration site selection. LEDGF/p75 consists of an integrase–binding domain (IBD), binding the viral IN, and a PWWP domain, binding epigenetic marks in the genome associated with active transcription such as H3K36me2/3. In addition, the three–dimensional organization in the nucleus and the epigenetic landscape of chromatin affect HIV integration, as HIV-1 tends to integrate into less condensed euchromatin and avoids integration in closed heterochromatin regions. (Figure created with Biorender.com on 17 December 2022).

**Figure 4 viruses-15-00032-f004:**
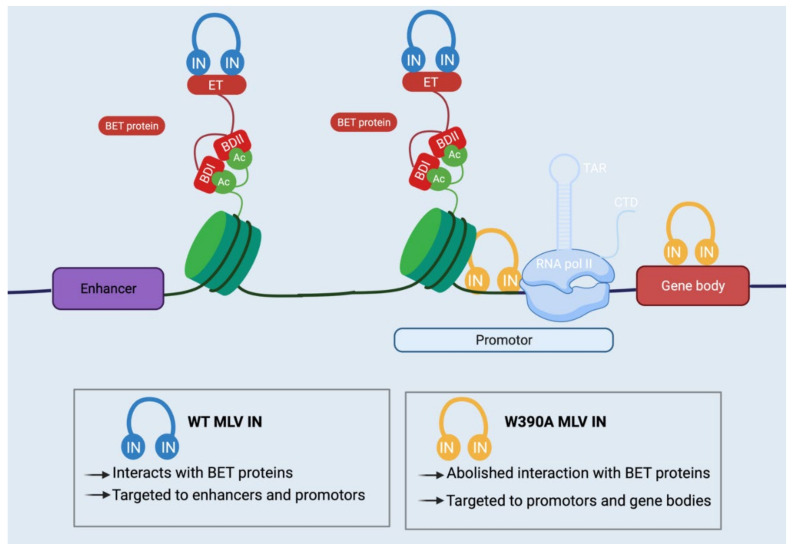
Integration pattern of wild–type (WT) and W380A mutant murine leukemia virus (MLV) in vivo. Bromodomain and extra–terminal motif (BET) proteins tether the wild–type (WT) MLV pre–integration complex (PIC) to enhancers and promoters. In contrast, integration of the W390A mutant MLV PIC, which cannot interact with BET, is redirected toward promoters and gene bodies. Oncogenesis by WT MLV IN is probably provoked by integration near enhancers and promotors, which results in the overexpression of cellular oncogenes, while the W390A mutant’s integration in oncogene bodies induces insertional mutagenesis [55] (Figure created with Biorender.com on 17 December 2022).

**Table 1 viruses-15-00032-t001:** Determinants of integration site selection of MLV and HIV-1. Bromodomain (BD); bromodomain and extra terminal motif (BET); catalytical core domain (CCD); cleavage and polyadenylation specificity factor 6 (CPSF6); C-terminal domain (CTD); extra-terminal domain (ET); hepatoma-derived growth factor-related protein 2 (HRP2); integrase-binding domain (IBD); integrase (IN); lens epithelium-derived growth factor (LEDGF/p75); nucleoporin 153 (Nup153); pre-integration complex (PIC); ran–binding protein 2 (RanBP2); transportin SR2 (TRN-SR2). (Table created with Biorender.com on 17 December 2022).

Host Factors and Viral Proteins Involved in Nuclear Import and/or MLV Integration Site Selection			
Host Factor	Viral Protein	Description	References
p12	/	Tether of PIC to mitotic chromosomes	[46,47,48,49,50]
BET	/	BD binds viral IN, ET binds chromatin	[28,29,30,31,32,33,34,51,52,53,54,55,56,57]
/	IN	CCD and CTD bind target DNA	[30,51,53,58,59]
**Host Factors and Viral Proteins Involved in Nuclear Import and/or HIV-1** **Integration Site Selection**			
**Host Factor**	**Viral Protein**	**Description**	**References**
TRN–SR2	/	Karyopherin involved in nuclear import	[60]
RanBP2	/	Nuclear pore protein involved in Nuclear import	[60]
Nup153	/	Karyopherin involved in nuclear import	[61,62]
CPSF6	/	Nuclear import factor binding to HIV-1 capsid	[63,64,65,66,67]
LEDGF/p75	/	IBD binds viral IN, PWWP domain binds H3K36me2/3	[22,23,35,36,37,38,39,40,41,42,45,52,68,69,70,71,72,73,74,75,76,77,78,79,80,81,82,83,84,85,86,87]
HRP2	/	IBD binds viral IN, PWWP domain binds H3K36me2/3	[74]
/	IN	CCD and CTD bind target DNA	[88,89,90,91,92,93,94,95,96,97,98]

**Table 2 viruses-15-00032-t002:** Determinants of retroviral integration site selection. Avian sarcoma-leukosis virus (ASLV); bromodomain and extra terminal motif (BET); bovine leukemia virus (BLV); facilitates chromatin transcription complex (FACT complex); feline immunodeficiency virus (FIV); group–specific antigen (Gag); human immunodeficiency virus type 1 (HIV-1); human immunodeficiency virus type 2 (HIV-2); human T lymphotropic virus type 1 (HTLV-1); human T lymphotropic virus type 2 (HTLV-2); integrase (IN); lens epithelium–derived growth factor (LEDGF/p75); mouse mammary tumor virus (MMTV); murine leukemia virus (MLV); protein phosphatase 2A (PP2A); Rous sarcoma virus (RSV); simian immunodeficiency virus (SIV). (Table created with Biorender.com on 17 December 2022).

Retroviral Integration Site Selection					
Retrovirus	Classification	Classification	Host Factor	Viral Protein	References
ASLV	*ortho-retroverinae*	alpharetrovirus	Unknown (FACT complex?)	/	[5,6,7,99,100,101]
RSV		alpharetrovirus	Unknown (FACT complex?)	IN	[8,102,103,104,105]
MMTV		betaretrovirus	/	/	[9,10,11,106,107,108]
MLV		gammaretrovirus	BET proteins	IN	see Table 1
HTLV-1		deltaretrovirus	PP2A	/	[13,109,110,111,112,113,114,115,116,117,118,119,120]
HTLV-2		deltaretrovirus	PP2A	/	[14,120]
BLV		deltaretrovirus	PP2A	/	[15,16,17,18,19,121]
HIV-1		lentiretrovirus	LEDGF/p75	IN	see Table 1
HIV-2		Lentiretrovirus	LEDGF/p75	/	[122,123,124,125,126,127]
SIV		Lentiretrovirus	LEDGF/p75	IN	[24,128,129]
FIV		lentiretrovirus	LEDGF/p75	/	[25,37]
Humanfoamy virus	*Spuma-retrovirinae*		/	Gag	[26,27,49,130,131,132,133]

## Data Availability

Not applicable.

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
