# Peer review of "Determinants of Retroviral Integration and Implications for Gene Therapeutic MLV—Based Vectors and for a Cure for HIV-1 Infection"

_viruses, 2022, doi:10.3390/v15010032_

Round 1

Reviewer 1 Report

The authors propose a review concerning retroviral integration and its implication in gene therapeutic and for HIV-1 Cure. The review is dealing with very “hot questions” and, thus, appears timely and very useful for the people working in the field.

The text is very well written and the figures readable and well designed. However, some points are not covered and could be added to increase the importance of the work. Anyway, I have only few comments regarding the current version:

-          Line 36 Authors indicate that cellular DNA repair enzymes trim the overhangs and fill the gaps after integration. Since the post-integration events and especially the DNA repair occurring after integration  remain mostly unknown I would chose “Cellular DNA repair enzyme are expected to trim…”

-          As well described in lines 219-226 the FACT has been only proved to regulate integration efficiency. Since its role in insertion site selection has not been yet demonstrated (using deep sequencing of the retroviral integration sites in FACT KO cells for example) I would  not mention it as host factor in the Table 2 related to integration site selection (or mention “unknown (FACT?)”.

-          In the description of the FACT also add the work reporting its role in facilitating HIV-1 integration into compacted chromatin (Matysiak et al, Retrovirology 2017).

-          Line 347, there is a typo in spelling Di Nunzio. Same typo in the bibliography part (lane 1265)

-          In the part describing the Medical applications for safer viral vectors some part of the literature is missing regarding punctual mutation within the IN protein (In the CTD DNA/chromatin binding domain) that affect and change the insertion site (see Benleulmi et al., Retrovirology 2017) and interaction with specific part of the chromosomes (Lapaillerie et al 2021). This could also be described in the Retroviral integration sites matter part of the review.

-          In the medical implication, it could be interesting also to mention the possible use of retroviruses and their insertion site preferences for Oncovirotherapy. Oncolytic Virotherapy (OVT) is an emerging therapy that uses replication-competent viruses to kill cancer cells. Various viruses have been applied in this purpose yielding encouraging results, including a retrovirus, Murine Leukemia Virus. The retroviral family Spumaretrovirinae comprising the zoonotic Prototype Foamy virus (PFV) exhibits properties that make it a promising oncolytic retrovirus. This includes no association with disease, viral genome integration profile favoring non-coding regions, replication limited to dividing cells, strong cytopathic effect and very low seroprevalence in the humans. A better understanding of the strategies developed by the virus to unsure efficient replication will be of great interest in order to engineer and optimize this retrovirus for future OVT applications. This could be also described in the review as an independent part or just mentioned in perspective.

Author Response

Point 1:  Line 36 Authors indicate that cellular DNA repair enzymes trim the overhangs and fill the gaps after integration. Since the post-integration events and especially the DNA repair occurring after integration remain mostly unknown I would chose “Cellular DNA repair enzyme are expected to trim…”

Response 1: Thank you for the comment. This is an interesting remark. We have corrected this sentence.

Point 2: As well described in lines 219-226 the FACT has been only proved to regulate integration efficiency. Since its role in insertion site selection has not been yet demonstrated (using deep sequencing of the retroviral integration sites in FACT KO cells for example) I would  not mention it as host factor in the Table 2 related to integration site selection (or mention “unknown (FACT?)”.As well described in lines 219-226 the FACT has been only proved to regulate integration efficiency. Since its role in insertion site selection has not been yet demonstrated (using deep sequencing of the retroviral integration sites in FACT KO cells for example) I would  not mention it as host factor in the Table 2 related to integration site selection (or mention “unknown (FACT?)”.

Response 2:  It is indeed only proven that the FACT complex regulates the integration efficiency, there I mentioned it as unknown in the table 2.

Point 3:  In the description of the FACT also add the work reporting its role in facilitating HIV-1 integration into compacted chromatin (Matysiak et al, Retrovirology 2017).

Response 3: This is indeed an interesting article to add to the review. The reference has been included.

Point 4: Line 347, there is a typo in spelling Di Nunzio. Same typo in the bibliography part (lane 1265)

Response 4: We have corrected the typo.

Point 5: In the part describing the Medical applications for safer viral vectors some part of the literature is missing regarding punctual mutation within the IN protein (In the CTD DNA/chromatin binding domain) that affect and change the insertion site (see Benleulmi et al., Retrovirology 2017) and interaction with specific part of the chromosomes (Lapaillerie et al 2021). This could also be described in the Retroviral integration sites matter part of the review.

Response 5: This was indeed missing. Both references are added to the review in the sections medical applications for viral vectors and integration sites matter.

Point 6: In the medical implication, it could be interesting also to mention the possible use of retroviruses and their insertion site preferences for Oncovirotherapy. Oncolytic Virotherapy (OVT) is an emerging therapy that uses replication-competent viruses to kill cancer cells. Various viruses have been applied in this purpose yielding encouraging results, including a retrovirus, Murine Leukemia Virus. The retroviral family Spumaretrovirinae comprising the zoonotic Prototype Foamy virus (PFV) exhibits properties that make it a promising oncolytic retrovirus. This includes no association with disease, viral genome integration profile favoring non-coding regions, replication limited to dividing cells, strong cytopathic effect and very low seroprevalence in the humans. A better understanding of the strategies developed by the virus to unsure efficient replication will be of great interest in order to engineer and optimize this retrovirus for future OVT applications. This could be also described in the review as an independent part or just mentioned in perspective.

Response 6: This is an interesting and constructive comment. An independent part about oncovirotherapy is added to the review.

Reviewer 2 Report

Eline Pellears et al, present a very detailed review of retrovirus integration and their application in gene-therapy. The paper provides detailed review of  several works highlighting integration preference of MLV, HIV-1 and other retroviruses. They also describe the involvement of host-factors in this process and provide some perspective to current findings. The clinical relevance of integration sites/ mechanisms is very clearly articulated in the text.

I have a one minor comments: Line 343: " HIV-1 tends to integrate into chromatin regions that are associated with nuclear pore complexes" I would think this is an old model, which has now been clearly demonstrated not to be true. The authors themselves point out that integration occurs >1um inside the nucleus (line 395). Perhaps providing a perspective to the afore mentioned interpretations will be useful.

In some places the authors argue that nuclear architecture plays a role in determining HIV-integration sites (line 467-470), but fail to mention more recent seminal papers that show virus targeting to nuclear compartments (speckles and Lamin). This information will add to a more even discussion of virus integration targeting in the context of the nuclear architecture. 

Author Response

Point 1: I have a one minor comments: Line 343: " HIV-1 tends to integrate into chromatin regions that are associated with nuclear pore complexes" I would think this is an old model, which has now been clearly demonstrated not to be true. The authors themselves point out that integration occurs >1um inside the nucleus (line 395). Perhaps providing a perspective to the afore mentioned interpretations will be useful.

Response 1: Thank you for your comment. This is indeed an older model, therefore I have adapted your comment in the new version of the review.

Point 2: In some places the authors argue that nuclear architecture plays a role in determining HIV-integration sites (line 467-470), but fail to mention more recent seminal papers that show virus targeting to nuclear compartments (speckles and Lamin). This information will add to a more even discussion of virus integration targeting in the context of the nuclear architecture. 

Response 2: Thank you for your comment. Papers about the speckles/lamin were indeed missing and references are added now to the review.

(Francis, A. C.; Marin, M.; Singh, P. K.; Achuthan, V.; Prellberg, M. J.; Palermino-Rowland, K.; Lan, S.; Tedbury, P. R.; Sarafianos, S. G.; Engelman, A. N.; Melikyan, G. B. HIV-1 Replication Complexes Accumulate in Nuclear Speckles and Integrate into Speckle-Associated Genomic Domains. Nat. Commun. 2020, 11 (1). https://doi.org/10.1038/s41467-020-17256-8.)

(Rensen, E.; Mueller, F.; Scoca, V.; Parmar, J. J.; Souque, P.; Zimmer, C.; Di Nunzio, F. Clustering and Reverse Transcription of HIV‐1 Genomes in Nuclear Niches of Macrophages. EMBO J. 2021, 40 (1), 1–16. https://doi.org/10.15252/embj.2020105247.)